# Video Instance Segmentation using Inter-Frame Communication Transformers

**Sukjun Hwang**[1]    **Miran Heo**[1]    **Seoung Wug Oh**[2]    **Seon Joo Kim**[1]
[1]Yonsei University    [2]Adobe Research
{sj.hwang, miran, seonjookim}@yonsei.ac.kr    seoh@adobe.com

## Abstract

We propose a novel end-to-end solution for video instance segmentation (VIS) based on transformers. Recently, the *per-clip* pipeline shows superior performance over *per-frame* methods leveraging richer information from multiple frames. However, previous per-clip models require heavy computation and memory usage to achieve frame-to-frame communications, limiting practicality. In this work, we propose Inter-frame Communication Transformers (IFC), which significantly reduces the overhead for information-passing between frames by efficiently encoding the context within the input clip. Specifically, we propose to utilize concise *memory tokens* as a means of conveying information as well as summarizing each frame scene. The features of each frame are enriched and correlated with other frames through exchange of information between the precisely encoded memory tokens. We validate our method on the latest benchmark sets and achieved state-of-the-art performance (AP 42.6 on YouTube-VIS 2019 val set using the offline inference) while having a considerably fast runtime (89.4 FPS). Our method can also be applied to near-online inference for processing a video in real-time with only a small delay. The code is available at https://github.com/sukjunhwang/IFC.

## 1 Introduction

With the growing interest toward the video domain in computer vision, the task of video instance segmentation (VIS) is emerging [1]. Most of the current approaches [1, 2, 3, 4] extend image instance segmentation models [5, 6, 7, 8] and take frame-wise inputs. These per-frame methods extend the concept of temporal tracking by matching frame-wise predictions of high similarities. The models can be easily customized to real-world applications as they run in an online [9] fashion, but they show limitations in dealing with occlusions and motion blur that are common in videos.

On the contrary, per-clip models are designed to overcome such challenges by incorporating multiple frames while sacrificing the efficiency. Previous per-clip approaches [10, 11, 12] aggregate information within a clip to generate instance-specific features. As the features are generated per instance, the number of instances in addition to the number of frames has a significant impact on the overall computation. Recently proposed VisTR [11] adapted DETR [13] to the VIS task and reduced the inference time by inserting the entire video, not a clip, to its offline end-to-end network. However, its full self-attention transformers [14] over the space-time inputs involve explosive computations and memories. In this work, we raise the following question: can a per-clip method be efficient while attaining great accuracy?

To achieve our goal, we introduce Inter-frame Communication Transformers (IFC) to greatly reduce the computations of the full space-time transformers. Similar to recent works [15, 16, 17] that alleviate the explosive computational growth inherent in attention-based models [14, 18], IFC takes a decomposition strategy utilizing two transformers. The first transformer (Encode-Receive, $\mathcal{E}$) encodes

each frame independently. To exchange the information between frames, the second transformer (Gather-Communicate, $\mathcal{G}$) executes attention between a small number of memory tokens that hold concise information of the clip. The memory tokens are utilized to store the overall context of the clip, for example *"a hand over a lizard"* in Fig. 1. The concise information assists detecting the lizard that is largely occluded by the hand in the first frame, without employing an expensive pixel-level attention over space and time. The memory tokens are only in charge of the communications between frames, and the features of each frame are enriched and correlated through the memory tokens.

We further reduce overheads while taking advantage of per-clip pipelines by concisely representing each instance with a unique convolutional weight [7]. Despite the changes of appearances at different frames, the instances of the same identity share commonalities because the frames originated from the same source video. Therefore, we can effectively capture instance-specific characteristics in a clip with dynamically generated convolutional weights. In companion with the segmentation, we track instances by uniformly applying the weights to all frames in a clip. Moreover, all executions of our spatial decoder are instance-agnostic except for the final layer which applies instance-specific weights. Accordingly, our model is highly efficient and also suitable for scenes with numerous instances.

In addition to the efficient modeling, we provide optimizations and an instance tracking algorithm that are designed to be VIS-centric. By the definition of $AP^{VIS}$, the VIS task [1] aims to maximize the objective similarity: space-time mask IoU. Inspired by previous works [13, 19, 20], our model is optimized to maximize the similarity between bipartitely matched pairs of ground truth masks and predicted masks. Furthermore, we again adopt the similarity maximization for tracking instances of same identities, which effectively links predicted space-time masks using bipartite matching. As both of our training and inference algorithms are fundamentally designed to address the key challenge of VIS task, our method attains an outstanding accuracy.

From these improvements, IFC sets the new state-of-the-art: 42.6% AP and more surprisingly, in 89.4 fps. Furthermore, our model also shows great speed-accuracy balance under near-online settings, which leads to a huge practicality. We believe that our model can be a powerful baseline for video instance segmentation approaches that follow the per-clip execution.

## 2    Related Work

**Video instance segmentation**    The VIS task [1] extends the concept of tracking to the image instance segmentation task. The early solutions [1, 2] follow the per-frame pipeline, which utilize additional tracking head to the models that are mainly designed to solve image instance segmentation. More advanced algorithms that are recently proposed [3, 4] take video characteristics into consideration, which result in improved performance.

Per-clip models [10, 11, 12] dedicate computations to extract information from multiple frames for higher accuracy. By exploiting multiple frames, per-clip models can effectively handle typical challenges in video, *i.e.*, motion blurs and occlusions. Our model is designed to be highly efficient while following the per-clip pipeline, which leads to fast and accurate predictions.

**Transformers**    Recently, transformers [14] are greatly impacting many tasks in computer vision. After the huge success of DETR [13], which has brought a new paradigm to the object detection task, numerous vision tasks are incorporating transformers [21, 22] in place of CNNs. For classification tasks in both NLP and computer vision, many adopt an extra classification token to the input of transformers [21, 23]. All the input tokens affect each other as the encoders are mainly composed of the self-attention, thus the classification token can be used to determine the class of the overall input. Similarly, DeiT [24] inserts an additional distillation token to transformers, and the novel usage leads to a higher data efficiency. MaX-DeepLab [20] adopted the concept of memory and proposed a novel dual-path transformer for the panoptic segmentation task [25]. By making use of numerous memory tokens to convey information, MaX-DeepLab integrates the transformer and the CNN by making both feedback itself and the other.

We further utilize the concept of the memory tokens to the videos. Using Inter-frame Communication Transformers, each frame runs independently while sharing their information with interim communications. The communications lead to higher accuracy while the execution independence between frames accelerates the inference.

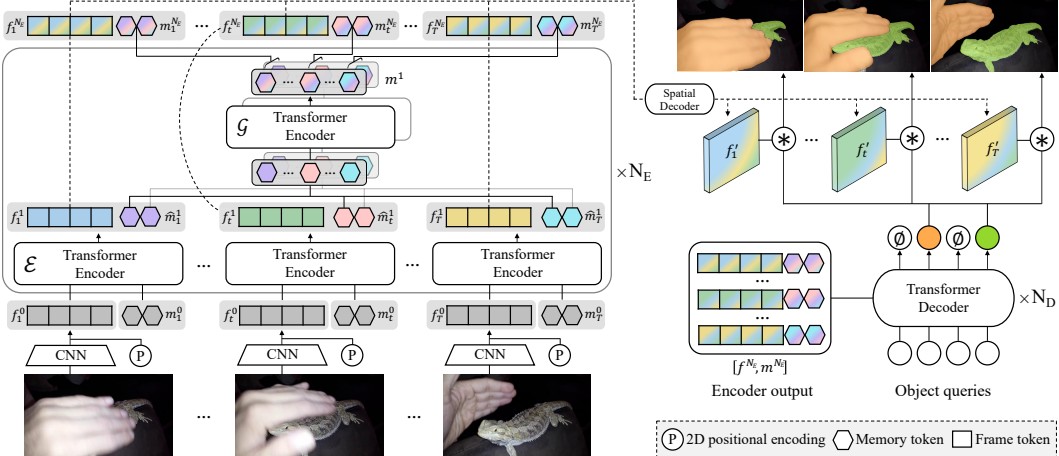

Figure 1: Overview of IFC framework. Our transformer encoder block has two components: 1) Encode-Receive ($\mathcal{E}$) simultaneously encodes frame tokens and memory tokens. 2) Only memory tokens pass Gather-Communicate ($\mathcal{G}$) to perform communications between frames. The output from the stack of $N_E$ encoder blocks goes into two modules, spatial decoder and transformer decoder, to generate segmentation masks.

## 3 Method

The proposed method follows a per-clip pipeline which takes a video clip as input and outputs clip-level results. We also introduce Inter-frame Communication Transformers, which can effectively share frame-wise information within a clip with a high efficiency.

### 3.1 Model architecture

Inspired by DETR [13], our network consists of a CNN backbone and transformer encoder-decoder layers (Fig. 1). The input clip is first independently embedded into a feature map through the backbone. Then, the embedded clip passes through our *inter-frame communication* encoder blocks that enrich the feature map by allowing information exchange between frames. Next, a set of transformer decoder layers that take the encoder outputs and object queries as inputs predict unique convolutional weights for each instance in the clip. Finally, the masks for each instance across the clip are computed in one shot by convolving the encoded feature map with the unique convolutional weight.

**Backbone**   Given an input clip $\{x_i\}_{i=1}^T \in \mathbb{R}^{T \times H_0 \times W_0 \times 3}$, composed of $T$ frames with 3 color channels, the CNN backbone processes the input clip frame-by-frame. As the result, the clip is encoded into a set of low-resolution features, $\{f_i^0\}_{i=1}^T \in \mathbb{R}^{T \times H \times W \times C}$, where $C$ is the number of channels and $H, W = \frac{H_0}{32}, \frac{W_0}{32}$.

**Inter-Frame Communication Encoder**   Given an image, humans can effortlessly summarize the scene with only a few words. Also, frames from a same video share a lot of commonalities, the difference between them is sufficiently summarized and communicated even with a small bandwidth. Based on this hypothesis, we propose an inter-frame communication encoder to make the computation to be mostly frame-wise independent with some communications between frames. Specifically, we adopt memory tokens for both summarizing per-frame scenes and the means of communications.

Our encoder blocks are composed of two phases of separate transformers: Encode-Receive ($\mathcal{E}$) and Gather-Communicate ($\mathcal{G}$). Both Encode-Receive and Gather-Communicate follow the typical transformer encoder architecture [14], which consists of an addition of fixed positional encoding, a multi-head self-attention module, and a feed forward network.

Encode-Receive operates in a per-frame manner, taking a frame-level feature map and corresponding memory tokens. Passing through Encode-Receive, we expect two functionalities: (1) image features encode per-frame information to the memory tokens, and (2) image features receive information of different frames that are gathered in the memory tokens. Gather-Communicate operates across frames

Table 1: Complexity comparison. Various transformer encoders for space-time input. As the overall FLOPs can vary by the number of detected instances, listed values are measured only at the encoders.

| Communication Type | Complexity per Layer | FLOPs (G)[1] | | | |
| --- | --- | --- | --- | --- | --- |
| | | $360 \times 640$ | | $720 \times 1280$ | |
| | | T=5 | T=36 | T=5 | T=36 |
| No Comm | $\mathcal{O}(C^2THW + CT(HW)^2)$ | 5.17 | 37.23 | 24.62 | 177.29 |
| Full THW | $\mathcal{O}(C^2THW + C(THW)^2)$ | 6.94 | 148.70 | 50.63 | 1815.38 |
| Decompose T-HW | $\mathcal{O}(C^2THW + CT(HW)^2 + CT^2HW)$ | 8.33 | 60.24 | 36.73 | 265.50 |
| **IFC** ($M = 8$) | $\mathcal{O}(C^2THW + CT(HW)^2)$ | 5.52 | 39.73 | 25.05 | 180.39 |

to form a clip-level knowledge. It takes the memory tokens from each frame as inputs and performs communications between frames. Alternating two phases through multiple layers, the encoder can efficiently learn consensus representations across frames.

In more detail, given the frame embedding $\{f_i^0\}_{i=1}^T$, we spatially flatten each feature $\mathbb{R}^{H \times W \times C} \to \mathbb{R}^{HW \times C}$. The initial memory tokens $m^0$ of size $M$ are copied per frame and concatenated to each frame feature as follows:

$$[f_t^0, m_t^0] \in \mathbb{R}^{(HW+M) \times C}, \qquad t \in \{1, 2, \cdots, T\}, \tag{1}$$

where $[\cdot, \cdot]$ indicates a concatenation of two feature vectors. Note that the initial memory tokens $m^0$ are trainable parameters learnt during training.

The first phase of IFC is Encode-Receive, which processes frames individually as follows:

$$[f_t^l, \widehat{m}_t^l] = \mathcal{E}^l([f_t^{l-1}, m_t^{l-1}]), \tag{2}$$

where $\mathcal{E}^l$ denotes the $l$-th Encode-Receive layer. With a self-attention computed over the frame pixel locations and the memory tokens, the information of each frame can be passed to the memory tokens and vise-versa.

The outputs of Encode-Receive are grouped by memory indices and formulate the inputs for Gather-Communicate layer. The grouping can be understood as a decomposition of memory tokens, and becomes computationally beneficial when the total size of gathered memory tokens increases.

$$[m_1^l(i), m_2^l(i), \cdots, m_T^l(i)] = \mathcal{G}^l([\widehat{m}_1^l(i), \widehat{m}_2^l(i), \cdots, \widehat{m}_T^l(i)]), \qquad i \in \{1, 2, \cdots, M\}, \tag{3}$$

where $\mathcal{G}^l$ denotes the $l$-th Gather-Communicate layer. The processed outputs are redistributed to the originated frame and get concatenated as $m_t = [m_t(1), m_t(2), \cdots, m_t(M)]$. Unlike Encode-Receive, Gather-Communicate utilizes the attention mechanism to convey the information from different frames over the input clip.

Defining the $l$-th inter-frame encoder block (IFC$^l$) as $\mathcal{E}^l$ followed by $\mathcal{G}^l$, the stack of $N_E$ encoder blocks can be inductively formulated as:

$$[f_t^l, m_t^l] = \text{IFC}^l([f_t^{l-1}, m_t^{l-1}]), \qquad 1 \leq l \leq N_E, \tag{4}$$

where $[f_t^{N_E}, m_t^{N_E}]$ is the final result. The stacking of multiple encoder layers brings communications between frames, thus each frame can have coincidence to the other, specifying the identities of instances in a given clip.

**Complexity comparison** In Table 1, we analyze the computational complexity of transformer encoder variants applied for video input in terms of the Big-O complexity and FLOPs. The complexity of the original transformer encoder layer [14] is $\mathcal{O}(C^2N + CN^2)$, where $N$ is the number of inputs. Without any communication between frames, *No Comm*, it shows the smallest amount of computation ($\mathcal{O}(C^2THW + CT(HW)^2)$). As indicated as *Full THW* in Table 1, the complexity of VisTR [11] that performs a full space-time self-attention is $\mathcal{O}(C^2(THW) + C(THW)^2)$ thus either a higher resolution or an increase of number of input frames leads to a massive increase in computations. VisTR bypasses the problem by highly reducing the input resolution and utilizing GPUs with tremendous

---

[1]Measured using `flop_count` function of `fvcore==0.1.5`.

memory capacity. However, as such solutions cannot resolve the fundamental issues, it is impractical to real-world videos. Moreover, VisTR remains as a complete offline strategy because it takes the entire video as an input.

An intriguing improvement for the naïve full self-attention would be the decomposition of the attention into space and time axis [16, 17, 26]. In *Decompose T-HW*, we decompose attention computation into spatial and temporal attention. The complexity of the separation of space-time leads to the sum of the two transformer encoder: $\mathcal{O}(T(C^2(HW) + C(HW)^2))$ and $\mathcal{O}(HW(C^2T + CT^2))$. In comparison to the full self-attention, the decomposition lowers the computational growth relative to the number of frames.

Our encoder, IFC, that communicates between frames using the memory tokens leads to a huge benefit to the total computations adding only a small amount of computation over *No Comm* while providing sufficient channels for communication. The complexity of each phase in our proposed encoder is: $\mathcal{O}(C^2T(HW + M) + CT(HW + M)^2)$ for Encode-Receive and $\mathcal{O}(C^2TM + CT^2M)$ for Gather-Communicate respectively. Assuming that $M$ is kept small (e.g., 8), the computation needed for Gather-Communicate can be neglected, while the complexity of Encode-Receive can be approximated to $\mathcal{O}(C^2THW + CT(HW)^2)$ as shown in Table 1. Finally, with respect to the number of frames of the input, we can expect approximate linear increase rather than the high increase of computation occurred in VisTR.

**Decoders and output heads**  As depicted in Fig. 1, the transformer decoder of our model is stacked with $N_D$ layers [14]. Contrary to VisTR, where the number of object queries increases proportionally to the number of frames, our model receives learnt encodings of fixed size $N_q$ for *object queries*. Also, by utilizing these encodings throughout the entire frames, our model can effectively deal with clips of various lengths. A set of projection matrices are applied to $\{f_t^{N_E}, m_t^{N_E}\}_{t=1}^T$ for the generation of keys and values. The object queries turn into *output embeddings* by the transformer decoder, and the embeddings are eventually used as an input to the output heads.

There are two output heads on top of the transformer decoder, a class head and a segmentation head, each composed of two fully-connected layers. The output embeddings from the transformer decoder are independently inserted to the heads, resulting in $N_q$ predictions per a clip. The class head outputs a class probability distribution of instances $\hat{p}(c) \in \mathbb{R}^{N_q \times |\mathbb{C}|}$. Note that the possible classes $\mathbb{C} \ni c$ include *no object* $\varnothing$ class in addition to the given classes of a dataset.

The segmentation head generates $N_q$ conditional convolutional weights $w \in \mathbb{R}^{N_q \times C}$ in a manner similar to [7, 20]. For the conditional convolution, the output feature of the encoder $\{f_t^{N_E}\}_{t=1}^T$ is reused by undoing the flatten operation. For the upsampling, the encoder feature passes through fpn-style [27] *spatial decoder* without temporal connections resulting in $T$ feature maps that are 1/8 of the input resolution. Finally, the resulting feature maps $f'$ are convolved with each convolutional weight to generate a segmentation mask as follows:

$$\hat{s}_i = \{f'_t \circ w_i\}_{t=1}^T, \tag{5}$$

where $w_i$ is $i$-th convolutional weight, $\circ$ indicate $1 \times 1$ spatial convolution operation, and the result $\hat{s}_i$ is a spatial-temporal object mask in shape of $\mathbb{R}^{T \times H' \times W'}$ where $H' = \frac{H_0}{8}$, $W' = \frac{W_0}{8}$. Note that, for an instance, a common weight is applied throughout the video clip. Our spatial decoder is an instance-agnostic design, which is much more efficient than instance-specific decoders [10, 11, 12, 13] as the number of detected instances increases. Meanwhile, thanks to our segmentation head which specifies and captures the characteristics of an instance, IFC can conduct both segmentation and tracking at once within a clip.

## 3.2  Instance matching and loss

To train our network, we first assign the ground truth for each instance estimation and then a set of loss function between each the ground truth and prediction pair. For a given input clip, our model generate a fixed-size set of class-labeled masks $\{\hat{y}_i\}_{i=1}^{N_q} = \{(\hat{p}_i(c), \hat{s}_i)\}_{i=1}^{N_q}$. The ground truth set of the clip can be represented as $y_i = (c_i, s_i)$; $c_i$ is the target class label including $\varnothing$, and $s_i$ is the target mask which is down-sampled to the size of the prediction masks for efficient similarity calculation. One-to-one bipartite matching between the prediction set $\{\hat{y}_i\}_{i=1}^{N_q}$ and the ground truth set $\{y_i\}_{i=1}^K$ is performed to find the best assignment of a prediction to a ground truth. The objective can be formally

described as:

$$\hat{\sigma} = \underset{\sigma \in \mathfrak{S}_{N_q}}{\arg\max} \sum_{i=1}^{K} \text{sim}(y_i, \hat{y}_{\sigma(i)}), \tag{6}$$

where $\text{sim}(y_i, \hat{y}_{\sigma(i)})$ refers a pair-wise similarity over a permutation of $\sigma \in \mathfrak{S}_{N_q}$. Following prior work [13, 20, 28], the bipartite matching is efficiently computed using Hungarian algorithm [19]. We find that box-based similarity measurement as used in DETR [13] shows weaknesses in matching instances in video clip due to the case of occlusion and disappear-and-reappear. Therefore, we define $\text{sim}(y_i, \hat{y}_{\sigma(i)})$ to be mask-based term as $\mathbb{1}_{\{c_i \neq \varnothing\}}[\hat{p}_{\sigma(i)}(c_i) + \lambda_0 \text{DICE}(s_i, \hat{s}_{\sigma(i)})]$, where DICE denotes dice coefficients [29].

Given the optimal assignment $\hat{\sigma}$, we refer to the $K$ matched predictions and $(N_q - K)$ non-matched predictions as positive and negative pairs respectively. The positive pairs aim to predict the ground truth masks and classes while the negative pairs are optimized to predict the $\varnothing$ class. The final loss is a sum of the losses from positive pairs and negative pairs where each can be computed as follows:

$$\mathcal{L}_{pos} = \sum_{i=1}^{K} [\underbrace{-\log \hat{p}_{\hat{\sigma}(i)}(c_i)}_{\text{Cross-entropy loss}} + \lambda_1 \underbrace{(1 - \text{DICE}(s_i, \hat{s}_{\hat{\sigma}(i)}))}_{\text{Dice loss [29]}} + \lambda_2 \underbrace{\text{FOCAL}(s_i, \hat{s}_{\hat{\sigma}(i)})}_{\text{Sigmoid-focal loss [30]}}],$$

$$\mathcal{L}_{neg} = \sum_{i=k+1}^{N_q} [-\log \hat{p}_{\hat{\sigma}(i)}(\varnothing)]. \tag{7}$$

As $(N_q - K)$ is likely to be much greater than $K$, we down-weight $\mathcal{L}_{neg}$ by a factor of 10 to resolve the imbalance, following prior work [13]. The goal of video instance segmentation [1] is to maximize the space-time IoU between a prediction and a ground truth mask. Therefore, our mask-related losses (Dice loss and Sigmoid-focal loss) are spatio-temporally calculated over an entire clip, rather than averaging the losses that are accumulated frame-by-frame.

### 3.3 Clip-level instance tracking

To infer a video input that is longer than the clip length, we match instances using the predicted masks of overlapping frames. Let $\mathcal{Y}_I$ and $\mathcal{Y}_A$ be the result sets of clip $I$ and $A$ excluding the $\varnothing$ class. The goal is to perform matching of same identities between pre-collected instance set $\mathcal{Y}_I$ and $\mathcal{Y}_A$. We first calculate the matching scores which are space-time soft IoU at intersecting frames between $\mathcal{Y}_I$ and $\mathcal{Y}_A$. Then, we find optimal paired indices $\hat{\sigma}_S$ using Hungarian algorithm [19] to the gathered matching score $\mathcal{S} \in [0, 1]^{|\mathcal{Y}_I| \times |\mathcal{Y}_A|}$. We update $\mathcal{Y}_I(i)$ by concatenating $\mathcal{Y}_A(\hat{\sigma}_S(i))$ if $\mathcal{S}(i, \hat{\sigma}_S(i))$ is above a certain threshold, and add non-matched prediction sets to $\mathcal{Y}_I$ as *new* instances. Note that a previous per-clip model (MaskProp [10]) also utilizes soft IoU for tracking instances, but the matching scores are computed per-frame and averaged for intersecting frames. Different from MaskProp, using space-time soft IoU leads to an accurate tracking as it can better represent the definition of mask similarities between clips which brings at most 2% AP increase. The overall tracking pipeline can be effectively implemented in a GPU-friendly manner.

## 4 Experiments

In this section, we evaluate the proposed method using YouTube-VIS 2019 and 2021 [1]. For every listed score, we report the mean of five runs as the results may vary by each run due to the insufficient number of training and testing set of YouTube-VIS dataset. We demonstrate the effectiveness of our model regarding both accuracy and speed. We further examine how different settings affect the overall performance and efficiency of IFC encoder. Unless specified, all models for measurements used $N_E = 3, N_D = 3$, stride of 1, and ResNet-50.

### 4.1 Implementation Details

We used `detectron2` [33] for our code basis, and hyper-parameters mostly follow the settings of DETR [13] unless specified. We used AdamW [34] optimizer with initial learning rate of $10^{-4}$ for transformers, and $10^{-5}$ for backbone. We first pre-train the model for image instance segmentation on COCO [35] by setting our model to $T = 1$. The pre-train procedure follows the shortened training

Table 2: Evaluations on various settings.

(a) AP and FPS on YouTube-VIS 2019 `val` set. For fairness, FPS is measured on a same machine, using a single RTX 2080Ti GPU. We used the official codes and checkpoints provided by the authors for the measurements. We report the clip settings of [10, 11]. $T$: window size.

| | Method (Settings) | | Backbone [31] | FPS[2] | AP | $AP_{50}$ | $AP_{75}$ | $AR_1$ | $AR_{10}$ |
|---|---|---|---|---|---|---|---|---|---|
| per-frame | MaskTrack R-CNN [1] | | ResNet-50 | 26.1 | 30.3 | 51.1 | 32.6 | 31.0 | 35.5 |
| | MaskTrack R-CNN [1] | | ResNet-101 | - | 31.8 | 53.0 | 33.6 | 33.2 | 37.6 |
| | SipMask [2] | | ResNet-50 | 35.5 | 33.7 | 54.1 | 35.8 | 35.4 | 40.1 |
| | SG-Net [4] | | ResNet-50 | - | 34.8 | 56.1 | 36.8 | 35.8 | 40.8 |
| | SG-Net [4] | | ResNet-101 | - | 36.3 | 57.1 | 39.6 | 35.9 | 43.0 |
| | CrossVIS [3] | | ResNet-50 | - | 36.3 | 56.8 | 38.9 | 35.6 | 40.7 |
| | CrossVIS [3] | | ResNet-101 | - | 36.6 | 57.3 | 39.7 | 36.0 | 42.0 |
| per-clip | STEm-Seg [32] | | ResNet-101 | 3.0 | 34.6 | 55.8 | 37.9 | 34.4 | 41.6 |
| | VisTR [11] | ($T$=36) | ResNet-50 | 51.1 | 35.6 | 56.8 | 37.0 | 35.2 | 40.2 |
| | VisTR [11] | ($T$=36) | ResNet-101 | 43.5 | 38.6 | 61.3 | 42.3 | 37.6 | 44.2 |
| | MaskProp [10] | ($T$=13) | ResNet-50 | - | 40.0 | - | 42.9 | - | - |
| | MaskProp [10] | ($T$=13) | ResNet-101 | - | 42.5 | - | 45.6 | - | - |
| | **Ours**$_{\text{near-online}}$ | ($T$=5) | ResNet-50 | 46.5 | 39.0 | 60.4 | 42.7 | 41.7 | 51.6 |
| | **Ours**$_{\text{offline}}$ | ($T$=36) | ResNet-50 | 107.1 | 41.2 | 65.1 | 44.6 | 42.3 | 49.6 |
| | **Ours**$_{\text{offline}}$ | ($T$=36) | ResNet-101 | 89.4 | 42.6 | 66.6 | 46.3 | 43.5 | 51.4 |

(b) Accuracy on YTVIS 2021 `val` set

| | AP | $AP_{50}$ | $AP_{75}$ |
|---|---|---|---|
| MaskTrack-RCNN | 28.6 | 48.9 | 29.6 |
| SipMask | 31.7 | 52.5 | 34.0 |
| CrossVIS | 34.2 | 54.4 | 37.9 |
| **Ours** | 35.2 | 57.2 | 37.5 |

(c) Bipartite matching

| | AP |
|---|---|
| Box-based | 37.5 |
| Mask-based | 39.6 |

(d) Effect of strides

| | | AP | $AP_{75}$ | FPS |
|---|---|---|---|---|
| T = 5 | S = 3 | 38.7 | 42.1 | 72.7 |
| T = 10 | S = 5 | 39.5 | 42.8 | 83.0 |
| T = 15 | S = 8 | 39.7 | 43.0 | 92.5 |
| T = 20 | S = 10 | 40.4 | 43.3 | 95.7 |

schedule of DETR [13], which runs 300 epochs with a decay of the learning rate by a factor of 10 at 200 epochs. Using the pre-trained weights, the models are trained on a targeted dataset using the batch size of 16, each clip composed of $T = 5$ frames downscaled to either 360p or 480p. For the sampling of each clip, a reference frame index $t$ is randomly chosen. The remaining $T - 1$ frame indices are then sampled within an interval of 20. The models are trained for 8 epochs, and decays the learning rate by 10 at 5th epoch.

During inference, our model takes inputs as follows. Let an input video has $V$ frames, $T$ is the number of frames per clip and $S$ is the stride of clips. We start from inserting a clip of frame indices $[1, T]$ and sequentially insert clips of $[1 + S, T + S], [1 + 2S, T + 2S], \cdots, [1 + nS, T + nS]$. It repeats until the end frame index $T + nS$ is equal to or greater than $V$. If the end frame index of the last clip $T + nS$ is greater than $V$, we change the frame indices of the last clip to $[V - T + 1, V]$. The resolution of input videos are downscaled to 360p, which follows MaskTrack R-CNN [1].

### 4.2 Main Results

**YouTube-VIS 2019 evaluation results** We compare our proposed IFC to the state-of-the-art models in the video instance segmentation task on YouTube-VIS 2019 `val` in Table 2 (a). We measure the accuracy by AP and our model sets the highest score among all online, near-online, and offline models while presenting the fastest runtime. As mentioned earlier, IFC is highly efficient during the inference thanks to three advantages: (1) memory token-based decomposition for transformer encoder (2) instance-agnostic spatial decoder (3) GPU-friendly instance matching. Moreover, our model does not make use of any heavy modules such as deformable convolutions [36] or cascading networks [37]. Thanks to these advantages, IFC achieves an outstanding runtime, which is faster speed than online models [1, 2].

During the inference, our method is able to freely adjust the length of the clip ($T$) as needed. If the input clip length is set to contain entire video frames, our method becomes an offline method (like

---

[2]We follow `detectron2` [33] for measuring FPS.

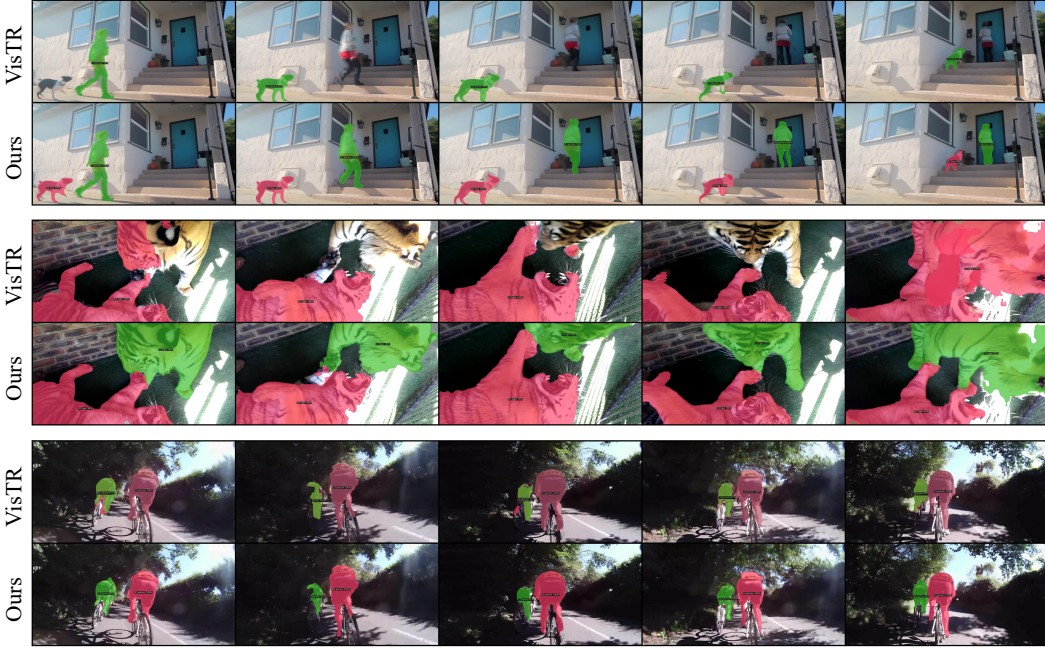

Figure 2: Visualization of predictions from VisTR and our model. Instances with the same identity are displayed in the same color.

VisTR [11]) that processes the entire video in one shot. As the offline inference can skip matching between clips and maximize the GPU utilization, our method represents surprisingly fast runtime (107.1 FPS). On the other hand, if the application requires instant outputs given a video stream, we can reduce the clip length to make our method near-online. In the near-online scenario with $T = 5$, our system is still able to process a video in real-time (46.5 FPS) with only a small delay.

**YouTube-VIS 2021 evaluation results**    The recently introduced dataset YouTube-VIS 2021 is an improved version of YouTube-VIS 2019. The newly added videos in the dataset include higher number of instances and frames. For the new dataset, we use 32 memory tokens. In Table 2 (b), we refer the results reported in [3], which evaluated [1, 2] using official implementations. Again, our model achieves the best performance.

**Qualitative result comparison**    We compare some qualitative results predicted by our model and VisTR [11] in Fig. 2. In terms of both tracking accuracy and segmentation quality, IFC yields better results than VisTR.

## 4.3    Ablation Study

In this section, we provide ablation studies and discuss how different settings impact the overall performance. The experiments are conducted using YouTube-VIS 2019 `val` set.

**Box-based and mask-based bipartite matching**    We observe how the different policies for bipartite matching affect the performance. As our model does is a box-free method, we adjust our model to predict bounding boxes similar to VisTR [11] and conduct bipartite matching [13, 19] using the predicted boxes. The change of optimization from mask-based to box-based brings a noticeable performance drop as shown in Table 2 (c). With the VIS-centric design, the mask-based optimization shows more robustness than box-based optimizations under typical video circumstances such as instances with heavy overlaps and partial occlusions.

**Differing window strides**    In addition to the clip length $T$, we further optimize our runtime placing a stride $S$ between clips, as shown in Table 2 (d). IFC can be used in a near-online manner, which takes clips that are consecutively extracted from a video. The placement of a larger stride reduces temporal intersections, which lessens computational overheads but also causes difficulty in matching

Table 3: Encoder variations. We show how different encoders affect the overall performance.

(a) Various encoders taking clips of different lengths (see Table 1)

| | T=5 | | | T=10 | | | T=15 | | | T=20 | | |
| | AP | $AP_{75}$ | FPS | AP | $AP_{75}$ | FPS | AP | $AP_{75}$ | FPS | AP | $AP_{75}$ | FPS |
|---|---|---|---|---|---|---|---|---|---|---|---|---|
| No Comm | 37.4 | 39.9 | 38.1 | 38.8 | 41.6 | 40.8 | 39.3 | 41.7 | 46.7 | 39.6 | 41.9 | 52.9 |
| Full THW | 37.2 | 40.0 | 37.6 | 38.8 | 41.2 | 35.5 | 39.8 | 42.6 | 32.9 | 39.7 | 42.8 | 34.8 |
| Decomp T-HW | 37.2 | 39.8 | 35.7 | 38.3 | 40.9 | 37.9 | 38.5 | 41.5 | 42.6 | 39.0 | 41.9 | 49.4 |
| IFC | 39.0 | 42.7 | 36.3 | 39.6 | 43.0 | 38.9 | 39.8 | 43.0 | 43.7 | 40.4 | 43.4 | 50.2 |

(b) Image instance segmentation on COCO `val` set

| | $AP^{COCO}$ | $AP_{50}^{COCO}$ |
|---|---|---|
| w/o mem | 35.0 | 56.6 |
| w/ mem | 35.1 | 56.5 |

(c) Number of memory tokens (AP)

| | T=5 | T=10 | T=15 | T=20 |
|---|---|---|---|---|
| M=1 | 37.6 | 39.2 | 39.4 | 39.4 |
| M=2 | 37.9 | 39.2 | 39.6 | 39.8 |
| M=4 | 38.0 | 39.5 | 39.7 | 39.9 |
| **M=8** | **39.0** | **39.6** | **39.8** | **40.4** |
| M=16 | 38.1 | 39.1 | 39.7 | 39.9 |

(d) Index-wise memory decomposition

| | T=5 | T=10 | T=15 | T=20 |
|---|---|---|---|---|
| Unified | 38.1 | 38.9 | 39.7 | 39.9 |
| **Decomp** | **39.0** | **39.6** | **39.8** | **40.4** |

instances. By enlarging the stride from $S = 1$ to $S = 3$, IFC accomplishes approximately 150% speed improvement with only 0.1% AP drop. The tendency of high speed gain and low accuracy drop persists under various conditions. Therefore, our model can be applied to conditions where the enlargement of strides is necessary, *i.e.*, using devices that are not powerful enough but has to maintain high inference speed.

**Various decomposition strategies of encoders**   In Table 1, we observed the computational gaps derived from the decomposition of the encoder layers. Extending Table 1, we now investigate the how the decomposition strategies affect the accuracy in Table 3.

The models are evaluated with variety of window sizes ($T = 5, 10, 15, 20$) as an increase of window size $T$ has pros and cons. When matching predictions from different clips, greater $T$ is advantageous due to an enlargement of temporal intersections between clips. On the contrary, frames in longer clips are likely to be composed of diverse appearances, which disrupt tracking and segmenting instances within a clip. Therefore, the key to the performance enhancement is to cope with the appearance changes by precisely encoding and correlating space-time inputs.

As shown in Table 3 (a), the *full* self-attention [11] surpasses the encoder without communications as the length of clips increase. However, the enlargement of the window size highly slows down the inference speed, and the improvements are marginal that the tremendous computation and memory usage cannot be compensated. The *decomposition of space-time* maintains comparable speed even if the window is large, but fails to achieve high accuracy.

Our model shows fast inference as the only additional computations of IFC are from utilizing a small number of memory tokens. Furthermore, by effectively encoding the space-time inputs with the communications between frames, IFC can take advantages of enlarging the window size, and surpasses other encoders.

**Memory tokens**   We also study the effects of utilizing memory tokens. As mentioned, the motivation of using the memory tokens is to build communications between frames. Different from the video instance segmentation task, the image segmentation task is consisted of a single frame. Therefore, the use of the memory tokens does not lead to improvements to the image instance segmentation task as mutual communications cannot be solely made (see Table 3 (b)). Meanwhile, the utilization of the memory tokens achieves great improvements by effectively passing the information between frames. Results in Table 3 (a, c) demonstrate that the use of memory tokens achieves higher accuracy than the encoder without any communications (*No comm*), which emphasizes the importance of the communications. We evaluate how the size of the memory tokens affect the overall accuracy in Table 3 (c) and set the default size of the tokens $M$ to be 8.

In Section 3.1, we demonstrated the formulation of the inputs for Gather-Communicate layer, which groups the outputs of Encode-Receive by memory indices. As aforementioned, the formulation can be considered as a decomposition of memory tokens: insertion to the Gather-Communicate layer by separate $M$ groups each consisting of $T$ tokens. In Table 3 (d), we investigate the impact of inserting

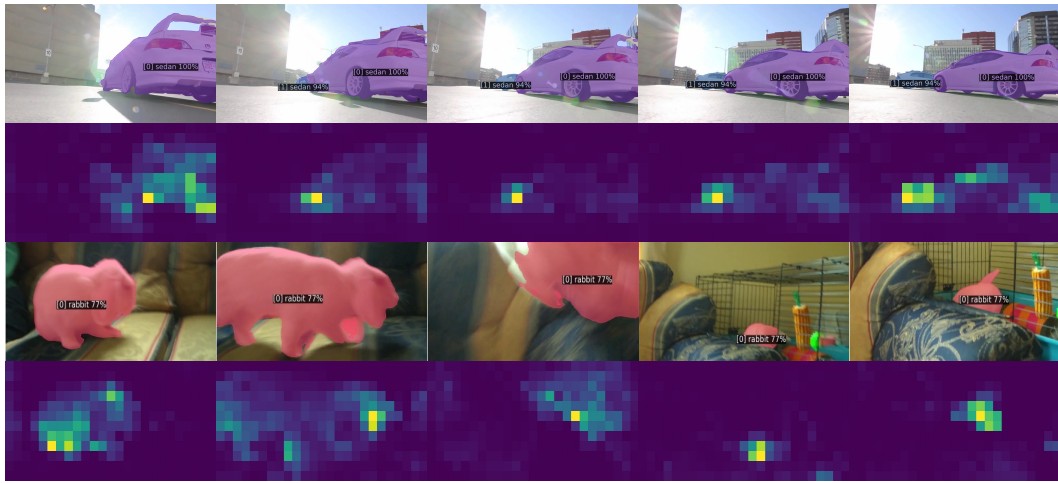

Figure 3: Visualizations of results and attention maps of memory tokens.

the unified $MT$ tokens as a whole. Compared to the *unified* insertion, the *decomposition* brings better accuracy as the memories of same indices have more correspondences, which ease the encoders to build attentions in between.

We choose a memory index attending foreground instances and visualize the attention map in Fig. 3. As shown in the results of the upper clip, we find that the memory token has more interests to instances that are relatively difficult to detect; it more attends the heavily occluded car at the rear. The clip at the bottom is composed of frames with huge motion blurs and appearance changes. With the communications of memory tokens, IFC successfully tracks and segments the rabbit.

## 5 Conclusion

In this paper, we have proposed a novel video instance segmentation network using Inter-frame Communication Transformers (IFC), which alleviates full space-time attention and successfully builds communications between frames. Finally, our network presents a rapid inference and sets the new state-of-the-art on the YouTube-VIS dataset. For the future work, we plan to integrate temporal information, which indeed would take a step further to the human video understanding.

## Acknowledgments

This research was grant funded by the Artificial Intelligence Graduate School Program of Yonsei University, under Grant 2020-0-01361, Korea Evaluation Institute of Industrial Technology (KEIT) funded by the Ministry of Trade, Industry and Energy (10073129), and also supported by the Advanced Robotics Laboratory, part of the Future Technology Center at LG Electronics.

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
