# OpenReview forum: "Video Instance Segmentation using Inter-Frame Communication Transformers"
_NeurIPS.cc/2021/Conference — NeurIPS 2021 Poster_

### Official Review · Reviewer_qqgE · 2021-07-14

**Rating:** 5
**Confidence:** 4

**Summary:**

- The paper proposes a video instance segmentation method based on Transformers. The authors propose to aggregate temporal features through memory tokens. Specifically,  each frame is summarized by several memory tokens.  A Transformer decoder is used to perform the inter-frame feature learning using these tokens.
- The paper shows strong performance on the Youtube-VIS benchmark, outperforming state-of-the-art methods in both speed and accuracy.

**Limitations And Societal Impact:**

The limitations are not discussed. The potential negative societal impact is provided.

**Main Review:**

# Strength
- Using memory tokens to perform inter-frame representation learning is highly motivated.
- The proposed method is highly efficient and effective, e.g., 46.5 fps/41.0 AP for near-online inference.

# Weakness
- Limited contributions. For each component in the proposed framework, only the memory token is the new technique. Others are proposed in previous papers, e.g., DETR and VisTR.
- According to the ablations, the biggest improvement (2.2 AP) is from the bipartite matching strategy, i.e., using mask-based matching instead of box-based one. It makes this work less impressive.
- The conclusion of 'we find that the memory token has more interests to instances that are relatively difficult to detect;' is not convincing. Are the visualized figures from only one of the memory tokens?  More analysis is needed to support this claim.
- The conditional convolutional weights, i.e., single-layer 1x1 conv, look more similar to SOLOv2 [A] instead of CondInst [7]. Related discussion is missed.
- Many typos and grammar errors, e.g.,  L137, L188, L291

[A] Wang et al. Solov2: Dynamic, faster and stronger.  In NeurIPS, 2020.


**Time Spent Reviewing:**

4 hours

---

> ### Author Response · Authors · 2021-08-10
> **Reply to Reviewer qqgE**
>
> We appreciate the reviewer for the thoughtful review and constructive feedback. We hope the following addresses all your concerns and questions.
>
> ---
>
> Q1: Concerns about limited contributions.
>
> A1: The core contribution of our paper is IFC, which significantly reduces the complexity of the encoder. In addition to the higher efficiency, using IFC leads to better performance. Furthermore, we explored and proposed various extensions of 2D to 3D as follows: efficient generation of clip-wise instance masks, using space-time masks for tracking and sequence matching. While some components of our model are proposed previous works, they have not yet been used in the VIS task or not regarded as a mainstream approach. Each architectural design choice of our model surely contributes to the high accuracy and efficiency.
>
> ---
>
> Q2: “The conclusion of 'we find that the memory token has more interests to instances that are relatively difficult to detect;' is not convincing. Are the visualized figures from only one of the memory tokens?”
>
> A2: All the visualized figures are from a fixed head of a fixed memory token (the visualization process is explained in Reviewer#pxiA-A6). The ground-truth labels of the YouTube-VIS testset are not publicly available. Therefore, we investigated several visualized heatmaps and came out with the qualitative analysis. Since it cannot be quantitatively proved, we will tone down our claim in the revision.
>
> ---
>
> Q3: Missing discussions regarding SOLOv2.
>
> A3: Thanks for your suggestion. We shall add discussions and cite SOLOv2.
>
> ---
>
> Q4: Typos and grammar errors.
>
> A4: Thanks for pointing out the typos and grammatical errors. We will thoroughly fix all mistakes in the revision.

---

### Official Review · Reviewer_tpaP · 2021-07-14

**Rating:** 3
**Confidence:** 5

**Summary:**

This paper proposes Inter-frame Communication Transformers (ICT) to solve the Video Instance Segmentation (VIS) task in one framework. In particular,  they propose to utiize concise memory tokens as a means of converying information as well as summarizing each frame content.  The method achieves the state-of-the-art performance  on YouTube-VIS 2019.

**Limitations And Societal Impact:**

The main issue of this paper is the novelty.  In particular, they just apply the core idea of MaxDeeplab [1] into video domain. Thus there are no fundamental new messages in case of feature or object query fusion.
Here are the details.

1.  Architecute and Loss : The end-to-end solution for VIS is firstly proposed by VisTR[3].  It has been proven effectively for predicting multiple queries across time. Moreover, the decoder head and loss are the same with original DETR[2] (Panoptic Segmentation) paper.  Thus in case of architecture and loss, this is not novel.

2. Method: The overall architecture is based on VISTR.  The core contribution of this paper is Inter-Frame Communication Encoder which contains Encode-Receive and Gather-Communicate. However, I can not tell the difference between the Query and Pixel Interaction in MaxDeeplab. They are same component by encoding the appreance information into Object Query via attention layers (while this paper call it token, also misleading and hard to under why called token).  Gather-Communicate process is most used in preovious video segmentation method in previous work by link encoded query across time [4].

In summary, there is no new concept or new message contribution. The overall contributions are  incremental and there are many similar components in related fields.  At least, it dose not reach the


There are other issues in case of experiment comparison and writing.
.
1. It is unfair to compare privous work including SG-Net and CrossVIS since most previous work only uses previous frame information. This paper and VisTR use the whole clip as inputs. Moreover, the FPS is also misleading since VisTR-framework process one clip each time rather than one image.

2.  What is pretrained model result on COCO since it has hudge effect on YouTube-VIS ?

3.  What is difference between token and query in DETR?


Reference:

[1] MaX-DeepLab: End-to-End Panoptic Segmentation with Mask Transformers CVPR-2021

[2] End-to-End Object Detection with Transformers ECCV-2020

[3] VISTR:  End-to-End Video Instance Segmentation with Transformers CVPR-2021

[4] Temporally Distributed Networks for Fast Video Semantic Segmentation CVPR-2020



**Main Review:**


Strength:

1, The overall writing is good.

2, The experiment results on YouTube-VIS are good.

3, The motivation is clear: reducing computation cost and of transformer and reducent information in video by mean of passing tokens.


**Time Spent Reviewing:**

6.5

---

> ### Author Response · Authors · 2021-08-10
> **Reply to Reviewer tpaP**
>
> We appreciate the reviewer for the thoughtful review and constructive feedback. Our answers for the questions are as follows.
>
> ---
>
> Q1: Concerns regarding the novelty.
>
> A1: We agree there exists many commonalities between our model and VisTR as both follow the transformer encoder-decoder structure proposed by DETR. VisTR utilizes the space-time full self-attention in the encoder, which results in more computational overhead as the resolutions and number of frames increase. Therefore, our major focus was to improve the encoder. IFC dramatically reduces the computation while achieving higher accuracy, thus bringing huge practicality.\
> The main purposes of the memory in IFC and Max-DeepLab [1] are different. The memory in Max-DeepLab is similar to Object Queries in DETR in that each is used to come out with a prediction of an instance. Therefore, there is no separate encoder-decoder structure in Max-DeepLab. In contrast, the memory tokens in IFC are not constrained to represent an instance. Rather, the tokens serve to convey information throughout the clip.\
> Following the concept of query-reference frame, APM [2] aggregates features from past frames and uses them for predicting the current frame. It means that if given a clip of length T, APM [2] should run T times in order to predict all T frames in the clip. In contrast, each frame taken into IFC simultaneously communicates and updates each other. Therefore, IFC can come up with the predictions of T frames with a single forward procedure.
>
> [1] MaX-DeepLab: End-to-End Panoptic Segmentation with Mask Transformers, CVPR 2021\
> [2] Temporally Distributed Networks for Fast Video Semantic Segmentation, CVPR 2020
>
> ---
>
> Q2: “It is unfair to compare privous work including SG-Net and CrossVIS since most previous work only uses previous frame information. This paper and VisTR use the whole clip as inputs.”
>
> A2: It is true that offline methods are more likely to be accurate than online methods. However, we believe comparing our work to both online and offline methods will be highly informative to the readers. In the revision, we shall further clarify online and offline in Table 2a.
>
> ---
>
> Q3: “The FPS is also misleading since VisTR-framework process one clip each time rather than one image.”
>
> A3: All FPS results were measured using the exact same environment and formula of (total number of frames) / (total time taken for processing).
>
> ---
>
> Q4: What is the pretrained model result on COCO?
>
> A4: The results on COCO are in Table 3b in the main paper. The result is about 35.1 AP.
>
> ---
>
> Q5: What is the difference between token and query in DETR?
>
> A5: The memory tokens in IFC do not specifically represent instances. Rather, they serve to convey information throughout the clip. In contrast, each object query in DETR is directly used for the predictions of an instance. Our model also has queries for the final predictions (‘Object queries’ in Figure 1).

---

### Official Review · Reviewer_1red · 2021-07-18

**Rating:** 5
**Confidence:** 4

**Summary:**

This paper proposes inter-frame communication transformers(IFC) for video instance segmentation. Compared to a full spatial-temporal transformer with complexity THW x THW, the proposed IFC module reduces it to THW + T(HW)^2 while preserving the temporal and spatial attentions. The proposed video instance segmentation network outputs a clip-level prediction, which is then stitched along the time stamps via Hungarian matching between the tracklet tubes. The experiments on YouTube-VIS 2019 dataset shows the effectiveness of the proposed method.

**Limitations And Societal Impact:**

The authors have adequately addressed the limitations and potential negative societal impact of their work.
For long-term tracking of the instances, please see the main review.

**Main Review:**

* The proposed memory tokens are memory-efficient and effective at learning temporal relationships between frames.

* The paper provides extensive experiments and ablation studies.

* The paper is overall well-written and easy to follow.

* It seems that the proposed method provides a good efficiency-accuracy trade-off. How does it compare with MaskProp [10]?

* If I understand it correctly, the span of instance tracking is controlled by T (which defaults to 5 time stamps). How does the proposed model handle long-term tracking of the instances (e.g. re-appeared objects after 5+ or T+ frames)? How much is the performance drop caused by missing such long-term tracking?

* Why is the reported score in Table 2 (41.0 AP) and in Table 3 (39.0) different? Is it that one is the best score and the other is average score out of multiple runs (5 runs)? I could not find the explanation for this gap, and this should be clarified.

* The proposed method looks similar to 'decomposed T + HW' attention. How is the 'T + HW' version implemented? and why is it worse than the proposed method?

* How is the 'full THW' implemented? It's opposed to an expectation that the 'full THW' model should be the upper bound in terms of accuracy. What is missing in the 'full THW' model, and what makes the proposed method perform better?

**Time Spent Reviewing:**

4 hours

---

> ### Author Response · Authors · 2021-08-10
> **Reply to Reviewer 1red**
>
> We appreciate the reviewer for the thoughtful review and constructive feedback. Our answers for the questions are as follows.
>
> ---
>
> Q1: “It seems that the proposed method provides a good efficiency-accuracy trade-off. How does it compare with MaskProp?”
>
> A1: There is no reported inference speed or publicly available official code of MaskProp. Therefore, the efficiency-accuracy trade-off of MaskProp could not be accurately measured. Considering the architectural design and many components [1, 2, 3, 4] that MaskProp utilizes, we infer that the upper bound of FPS is around 5.
>
> [1] Hybrid task cascade for instance segmentation, CVPR 2019\
> [2] Aggregated residual transformations for deep neural networks, CVPR 2017\
> [3] Object detection in video with spatiotemporal sampling networks, ECCV 2018\
> [4] Deformable convolutional networks, ICCV 2017
>
> ---
>
> Q2: “How does the proposed model handle long-term tracking of the instances? How much is the performance drop caused by missing such long-term tracking?”
>
> A2: As the reviewer pointed out, the span of instance tracking is controlled by the window size T. Therefore, using a bigger window size leads to the higher chance of successfully dealing with instances that reappear after a long time interval. Due to the ground-truth of YouTube-VIS testset is not publicly available, we could not quantitatively measure the performance drop.
>
> ---
>
> Q3: “Why is the reported score in Table 2 and in Table 3 different?”
>
> A3: Sorry for the confusion. Due to the insufficient number of test videos in YouTube-VIS dataset, the scores may vary by each run. To provide a better analysis, the ablation studies (Table 3a, 3c, and 3d) are filled with mean values of five runs. Meanwhile the SOTA comparison (Table 2a) reports the max score of the five runs. We will further specify this in the revision.
>
> ---
>
> Q4: How is 'T + HW' version implemented?
>
> A4: A ‘T + HW’ transformer encoder block is composed as follows. Firstly, the frame features pass through a spatial transformer encoder (among frame features sharing the same frame index). Secondly, a temporal transformer encoder is applied (among frame features sharing the same spatial index).
>
> ---
>
> Q5: How is 'Full THW' implemented?
>
> A5: A ‘Full THW’ transformer encoder receives all frame features in a clip at once, which is the exact same as that of VisTR.
>
> ---
>
> Q6: Why does IFC perform better than ‘T + HW’ and ‘Full THW’?
>
> A6: While temporal communication is built only between the same spatial indices in ‘T+HW’, the memory tokens of IFC have a spatially global receptive field. The flexibility of IFC can ease dynamical approaches to relevant information and be beneficial for tracking instances.\
> Since ‘Full THW’ involves more features, the accuracy of ‘Full THW’ could be expected to be the upper bound. However, the more information can result in confusing the model of finding relevant information. In other words, concisely encoding information and simplifying the routes of accessing information can help models.

---

### Official Review · Reviewer_pxiA · 2021-07-26

**Rating:** 5
**Confidence:** 5

**Summary:**

The paper extends transformer-based video instance segmentation framework VisTR in three aspects. First, the intra- (i.e., space) and inter-frame (i.e., time) relation computations in the transformer encoder are decomposed. Specifically, the extra memory tokens are introduced for efficient intra- and inter-frame communication. Second, the fixed object queries are used in the transformer decoder and thus the model can take a varying number of input frames. Moreover, the contextualized object queries are forwarded to the segmentation head so that the adaptive convolutional weights can be generated for effective tracking and segmentation. Third, the authors presented to use 3D IoU to link the already tracked tracklets and the currently generated tracklets to produce final video-level tracklets. Combining all together, the final framework is evaluated on the benchmarks and achieves new state-of-the-art.

**Main Review:**

[Originality]

The main contribution of this paper lies in presenting a solid transformer-based video instance segmentation framework that pushes the state-of-the-art result significantly and is fast. The idea of introducing memory tokens for effective intra- and inter-frame relation computation is novel and interesting. ​
However, there are also several ideas that are not technically new.

- The similar memory-token-based relation computation is also described in the following work. However, I will not count the following paper as a concurrent work since it is published in arXiv near the submission. Please compare the proposal with the following work.

[1] End-to-End Video Object Detection with Spatial-Temporal Transformers, arXiv 2021

- The fixed object queries which are shared across the frame are also tried in the original VisTR paper (please refer to the original paper Table-2).
- The idea of using adaptive convolutional weights for instance segmentation is a promising technique that is verified in various previous literature (e.g., CondInst, SOLO, Max-deeplab).
- Using an FPN-like spatial decoder obviously gives an extra performance gain in pixel-level prediction tasks.
- The mask-based sequence matching is a direct extension of VisTR's box-based matching. While clearly effective, it consumes more memory than the box-based counterpart.
- The final loss functions (i.e., dice loss + focal loss) are from the VisTR.
- The 3D IoU score for clip-linking is a direct video-level extension of MaskProp's averaged 2D IoU scores.

[Quality]

- The authors well-combined the previous techniques into a single framework.
- Extensive ablation studies show the efficacy of the proposals.
- The video results are quite impressive.

[Clarity]

- As the key technical contribution is on the memory token, please discuss and compare it with the recent space-time factorization methods in more detail. For example, the following paper also introduces space-time factorization methods for video embeddings in the transformer.

[1] ViViT: A Video Vision Transformer, arXiv 2021

[2] Is Space-Time Attention All You Need for Video Understanding?, ICML 2021

- The method section is well-written and easy to follow. Table-1 shows a clear advantage of the proposal over the baselines.

- Typos
[1] Line 291 Our model is shows -> Our model shows

[Significance]
- While there are several components that are not new, each component is combined in a novel way and the final model achieves non-trivial gain over the baselines both in accuracy and speed.
- As transformer-based video models are becoming more and more important these days, the work provided has several strong points (memory token-based transformer encoder design and efficient tracklet matching) that need to be shared with the community.

[Question & Suggestion]
- The position encoding remains to be the same as VisTR. Did the authors have tried to drop it or try different types of embeddings? For example, the position prior can be embedded in the self-attention operation such as Swin Transformer.
- Why do more memory tokens fail to provide performance gain (e.g., 8 vs. 16 in Table 3-(c))? More discussions need to be provided in the manuscript.
- Please detail the visualization process in Figure 2.
- Please provide a full model FLOPS comparison by adding an extra column in Table 3.
- The authors can try to make the whole framework purely based on the Transformer (e.g., replacing the ResNet backbone with SETR).

**Needs Ethics Review:**

Yes

**Time Spent Reviewing:**

5 hours

---

> ### Author Response · Authors · 2021-08-10
> **Reply to Reviewer pixA**
>
> We appreciate the reviewer for the thoughtful review and constructive feedback. Our answers for the questions are as follows.
>
> ---
>
> Q1: “Please compare the proposal with the following work [1].”
>
> A1: We appreciate the reviewer for sharing the recent work, TransVOD [1].
> The main architectural design of TransVOD is based on the concept of ‘query-reference’. Given a clip, information from reference frames are aggregated using deformable attention for improving the predictions made at the current frame (query). In order to come up with predictions for all frames in a clip of length T, TransVOD must run T times.
> In contrast, all frames taken into IFC are treated equally. IFC makes frame features and memory tokens alternately interact and update each other. This aspect of IFC leads to high efficiency when generating predictions for all frames in each clip, which is necessary for tracking instances using space-time soft IoU.
>
> [1] End-to-End Video Object Detection with Spatial-Temporal Transformers, arXiv 2021
>
> ---
>
> Q2: There are several ideas that are not technically new.
>
> A2: Since the core contribution of our paper is IFC, we agree that there are several components that have been used from previous works (e.g., adaptive convolutional weights and FPN-like spatial decoder). However, the components were either remaining underrated of its potential or have not been explored for video instance segmentation. Each network design choices and proposed extensions from 2D to 3D (e.g., space-time instance segmentation, sequence matching and clip-linking) greatly improves overall accuracy and efficiency.
>
> ---
>
> Q3: Comparison with the recent space-time factorization methods.
>
> A3: As the reviewer mentioned, there has been recent works regarding the space-time factorization methods [1, 2].\
> ViViT [1] and TimeSformer [2] have proposed various decomposition methods to naïve full self-attention over space-time. The decompositions show promising results while effectively reducing overall computations.\
> TimeSformer and Model 3 of ViViT paper have two self-attentions that are computed independently from each other: spatial and temporal self-attentions. The spatial self-attention is over all tokens having the same temporal index, and vice-versa for the temporal self-attention. Iteratively switching the two self-attentions makes encoding space-time information possible. This strategy is similar to Axial-DeepLab [3] decomposing 2D self-attention into two 1D self-attentions, and also to R(2+1)D [4] decomposing C3D [5] into space and time.\
> Model 2 of ViViT paper is similar to IFC in that it accumulates spatial information into a single frame-level representation token. As effectively aggregating the global context is the key to the success of classification tasks, the use of a single token could be effective in the video classification task.
> Compared to the video classification task, the video instance segmentation (VIS) task requires more localized information in order to cope with instance relations. Furthermore, the frame features should be updated using aggregated information for better segmentation predictions. Therefore, IFC utilizes multiple memory tokens to encode frames which is more effective than using a single token (Table 3c). Moreover, frame features and memory tokens alternately interact and update each other in IFC, which lead to high segmentation results.
> In the revision, we shall add these comparisons and discussions. We appreciate your suggestion.
>
> [1] ViViT: A Video Vision Transformer, arXiv 2021\
> [2] Is Space-Time Attention All You Need for Video Understanding?, ICML 2021\
> [3] Axial-DeepLab: Stand-Alone Axial-Attention for Panoptic Segmentation, ECCV 2020\
> [4] A Closer Look at Spatiotemporal Convolutions for Action Recognition, CVPR 2018\
> [5] Learning Spatiotemporal Features with 3d Convolutional Networks, ICCV 2015
>
> ---
>
> Q4: Different types of positional embeddings.
>
> A4: We tried two positional embeddings to frame features: 3D spatio-temporal encoding of VisTR, and 2D spatial encoding of DETR. We also tested the effects of adding temporal embeddings to the memory tokens. Among the variations, using spatial encodings for the frame features and keeping memory tokens without the temporal embeddings showed highest accuracy. We will more specify about the embeddings in the revision.
>
> ---
>
> Q5: Why using 8 tokens shows the highest accuracy?
>
> A5: We do not believe 8 is the optimal number for all circumstances. For example, when handling complex scenes where multiple instances are involved, adopting more memory tokens would be helpful. In contrast, specifying relevant memory tokens would become easier with a smaller number. So, the smaller number of tokens might come up with better results for relatively monotonic scenes.
>
> ---
>
> Q6: Details of the visualization process in Figure 2.
>
> A6: We first collected the output of the last Encode-Receive module and visualized the attention scores of each head of each memory token. From the visualized outputs, we discovered that there constantly exists a head of a certain memory token which shows interests to foreground instances. We then fixed the head for visualizing all qualitative results (figures and videos in the main paper and supplementary).
>
> ---
>
> Q7: A full model FLOPS comparison and pure-transformer framework.
>
> A7: Thanks for your suggestions. We shall consider them in the revision.
>
> ---
>
> Q8: Typos
>
> A8: We thank the reviewer for pointing out the typo. We will make sure to fix all typos in the revision.

---

### Review · Ethics_Reviewer_4SyT · 2021-08-09

**Recommendation:** N/a

**Ethics Review:**

I believe this paper was mistakenly flagged for ethics review by pxiA

---

### Review · Ethics_Reviewer_QZwH · 2021-08-12

**Recommendation:** N/A

**Ethics Review:**

I'm not sure why this was tagged for ethical review. I suppose there are long-term implications for surveillance that could be problematic. But, it's a bit of a long walk to get there from this paper.

---

### Decision · Program_Chairs · 2021-09-28

**Decision:**

Accept (Poster)

**Comment:**

None of the reviewers recommended accepting this paper.
After reading the author response and other reviews one of the reviewers also reduced their score.
One of the common critiques of the work was around the degree of novelty provided by the work.
One of the initially more positive reviewers did recognize that the memory-token based inter-frame attention is technically new and felt the experiments here were quite extensive. But this reviewer also felt that the main performance improvement in this method were coming from the other elements of the method.

The AC recommends that this paper be rejected.


**Consistency Experiment:**

NeurIPS has a long history of experimentation. In 2014, NeurIPS ran an experiment in which 10% of submissions were reviewed by two independent committees to quantify the randomness in the review process. This year, we repeated a variant of this experiment to see how the quality of the review process has changed over time.  This paper was part of the experiment and was therefore assigned to two committees (consisting of reviewers, an Area Chair, and a Senior Area Chair) that reached independent decisions.  If both committees made the same recommendation, this recommendation was followed. If a single committee recommended acceptance, the paper was accepted (with the exception of a few cases in which the other committee identified what we considered a fatal flaw, e.g., an error in a key result).

This copy’s committee reached the following decision: **Reject**

The other committee assigned to the paper recommended **Accept (Poster)**.  You can find the other set of reviews, along with any follow up discussion with the authors here:
https://openreview.net/forum?id=s95BePNvykX